# Impact of Ultrasound on a Gluten-Free Composite Flour Based on Rice Flour and Corn Starch for Breadmaking Applications

**DOI:** 10.3390/foods14071094

**Published:** 2025-03-21

**Authors:** Mahsa Farrokhi, Ines N. Ramos, Cristina L. M. Silva

**Affiliations:** CBQF—Centro de Biotecnologia e Química Fina—Laboratório Associado, Escola Superior de Biotecnologia, Universidade Católica Portuguesa, Rua Diogo Botelho 1327, 4169-005 Porto, Portugal; mfarrokhi@ucp.pt (M.F.); iramos@ucp.pt (I.N.R.)

**Keywords:** ultrasound treatment, composite gluten-free flour, techno-functional properties, thermal analysis, rheological properties, structural modifications

## Abstract

Ultrasound (US) treatment is an eco-friendly physical modification technique increasingly used to enhance the functionality of gluten-free flours. In this study, the impact of sonication on the techno-functional, thermal, structural, and rheological properties of a composite gluten-free flour was investigated. The flour, comprising corn starch, rice flour, and other ingredients, was treated at hydration levels of 15% and 25% (*w*/*w*) under controlled conditions (10 min of sonication at 20 °C) and compared to a non-sonicated control. Sonication reduced the water absorption capacity (WAC) and swelling power (SP) while increasing the oil absorption capacity (OAC) and water solubility (WSI). Thermal analysis revealed lower gelatinization enthalpy, indicating structural modifications induced by cavitation. Structural assessments (XRD and FTIR) confirmed minimal alterations in crystallinity and short-range order. Rheological studies demonstrated an enhanced elasticity in the gel structure, especially at 15% hydration, while a morphological analysis via SEM highlighted particle fragmentation and surface roughening. These findings demonstrate the potential of ultrasound to modify gluten-free flours for improved functionality and diverse applications in gluten-free product development.

## 1. Introduction

Gluten, a protein found in wheat, barley, and rye, functions like a binding agent, providing dough with its distinctive elasticity [1]. However, for some individuals, gluten triggers an intolerance in the digestive system, requiring them to permanently avoid gluten-containing foods. The global demand for gluten-free products grew by approximately 16% between 2018 and 2022, making them one of the top 10 food trends today [2]. To meet the demand for high-quality gluten-free products, continuous advancements in research and development are essential. Formulating and optimizing gluten-free systems requires analyzing factors like texture, nutritional content, and sensory qualities to ensure high product quality and functionality [3].

Among all the gluten-free products, bread remains one of the most challenging to produce without gluten. This presents significant difficulties for bakers and cereal researchers, as gluten plays a critical role in dough rheology, processing, and final product quality. Gluten proteins are essential for forming a cohesive network that retains carbon dioxide during fermentation, enabling proper dough expansion and structure development [4]. In terms of consumer preferences, Alencar et al. [5] found that 93% of Brazilians with a gluten intolerance consume gluten-free bread (GFB) daily, but the texture and taste are the main reasons for their dissatisfaction with the available products. Many studies continue to highlight the desire for GFB with an improved texture and flavor, along with a greater variety [5,6,7]. Therefore, factors such as ingredients, nutritional value, and a wider selection of bread are critical when addressing the needs of consumers with gluten-related disorders, including celiac disease [5,8]. As a result, gluten-free bread often falls short in quality compared to gluten-containing bread.

Arendt and Bello [9] suggested using composite gluten-free flours, a blend of different gluten-free flours, rather than relying on a single flour type to produce GFB with better sensory and textural properties. They also noted that incorporating a certain percentage of starch can enhance the overall quality of gluten-free bread. Mixtures of gluten-free flours and starches, such as corn starch and rice flour, are commonly used [10]. However, there has been growing interest in other gluten-free flours, like teff, sorghum, millet, buckwheat, quinoa, and chickpea. Pre-gelatinized starches, produced by heating in the presence of water, are widely used for their functional properties, such as solubility in hot or cold water, high viscosity, and smooth texture. These starches are valuable in food processing where thickening is required [11]. Tsai et al. [12] also confirmed that adding rice slurry to bread dough improved the texture by promoting a softer crumb.

When developing gluten-free products, it is important to consider the functional properties of flour. Functional properties refer to the physicochemical characteristics that reflect the interaction between food components, their structure, and the environment in which they are measured [13]. Understanding these properties provides insight into how proteins, fats, fibers, and carbohydrates behave in specific food systems, aiding in their selection for various applications without compromising taste or texture [14]. Traditionally, the industry has approached the use of native flours, without modification, for the development of gluten-free products, with their properties depending on flour characteristics and composition, as well as the milling system used. However, flours produced by traditional methods can be subjected to various modification treatments, which can alter flour functionality and their suitability for different gluten-free formulations [15].

The food industry employs various modification techniques, such as genetic, mechanical, physical, chemical, or enzymatic methods, to alter the natural physicochemical properties of gluten-free ingredients, enabling them to better meet specific processing needs [16,17]. Modifying raw GF materials before dough preparation is crucial for improving bread quality, as starch granule characteristics significantly impact the final product due to their interactions with other components in the bread formulation [18]. Among the different modification techniques, physical processes like ultrasonication (US), high hydrostatic pressure (HHP), microwave (MW), extrusion, ozone treatment, drum drying (DD), heat-moisture treatment (HMT), and annealing (ANN) are favored in the food industry for their environmental benefits and ability to produce ’clean label’ products.

Ultrasound is an innovative physical processing method that has the potential to modify the functional properties of flours, leading to improvements in the rheological behavior of gluten-free dough and bread, as well as enhancing nutritional properties such as starch digestibility in gluten-free products [19]. Previous studies have demonstrated its impact on water absorption, solubility, swelling behavior, and rheological characteristics, making it a useful tool for improving gluten-free formulations [20]. Compared to conventional methods such as enzymatic or hydrothermal treatments, ultrasound offers a sustainable, non-thermal alternative with minimal chemical alterations, making it a green, safe, and easily applicable technology that has gained significant attention [21,22]. Acoustic cavitation, caused by the effect of ultrasonication, can modify the composition of the material. Modification caused by the cavitation effect can lead to the rapid formation, growth, and the collapse of bubbles in ultrasonic media, which causes mechanical bond fracture and free radical oxidation, thereby altering the structure of compounds and ultimately changing their composition [23].

This study explores the impact of ultrasound treatment on composite gluten-free flour at two different flour concentrations (15% and 25% (*w*/*w*)), addressing a gap in research regarding ultrasound’s application for modifying composite gluten-free flour. The investigation examines changes in techno-functional, thermal, rheological, structural, and morphological properties, comparing sonicated to non-sonicated (control) samples to evaluate the potential of ultrasound as a technique for enhancing flour quality.

## 2. Materials and Methods

### 2.1. Materials

The gluten-free composite flour was kindly provided by Germen—Moagem de Cereais S.A (Porto, Portugal). This blended flour consists primarily of corn starch and rice flour (49.11% and 23.86%, respectively), along with other ingredients like pregelatinized rice flour (8.58%), potato starch (5.6%), dextrose (4.9%), chickpea flour (2.2%), potato protein (2%), salt (1.5%), xanthan gum (1%), HPMC (Hydroxypropyl Methyl Cellulose) (1%), transglutaminase (0.2%) and carob flour (0.05%). Nutritionally, it is low in fat (0.4 g/100 g) and sugar (4.7 g/100 g) and high in fiber (6.2 g/100 g).

### 2.2. Ultrasound Treatment

Flour modification was carried out using a Qsonica Sonicator ultrasonic processor (Model Q700, Newtown, CT, USA) with a 25.4 mm diameter probe. The sonicator functioned at 20 kHz with an 80% amplitude and a maximum power output of 96 W. The process was conducted at a temperature of 20 °C, regulated by water circulation within a double-walled glass chamber and continuous mixing to prevent the sedimentation of the dispersion. Flour dispersions, prepared at concentrations of 15% and 25% (*w*/*w*) (total amount of 400 mL), were pre-mixed with a magnetic stirrer for 30 min to achieve uniformity before sonication. The sonication treatment was performed for 10 min, after which the mixtures were placed into tubes and stored at −18 °C for 24 h. To recover the sonicated flours, the frozen dispersions were freeze-dried (Armfield SB4 model, Ringwood, UK), ground, and sieved to <250 µm to obtain uniform particle sizes. The moisture content of the recovered flour was then adjusted to 10–12% for further analysis. These sonication conditions were selected based on the study by Farrokhi et al. [24] which reviewed and compared the effects of various sonication conditions on flours and starches. The most commonly used and effective conditions reported in the literature were chosen for this research. Samples were identified as non-sonicated (control, without sonication), 15%-sonication (sonicated with 15% (*w*/*w*) flour concentration), and 25%-sonication (sonicated with 25% (*w*/*w*) flour concentration).

### 2.3. Functional Properties

The functional properties of the flour samples, including water absorption capacity (WAC), oil absorption capacity (OAC), water absorption index (WAI), water solubility index (WSI), and swelling power (SP), were measured. Each test was conducted in six replicates.

Water absorption capacity (WAC) was evaluated using a modified method of Sosulski [25]. A 3 g flour sample was thoroughly mixed with 25 mL of distilled water in a 50 mL centrifuge tube. After allowing the mixture to rest for 30 min, it was centrifuged at 3000× *g* for 30 min at 4 °C. The supernatant was then decanted, and the sediment was dried in an oven (VWR^®^ DRY-Line^®^) at 50 °C for 25 min. WAC was calculated by subtracting the weight of the dried sediment from the initial mass of the flour dispersion and then dividing it by the weight of the flour sample. The WAC was expressed as grams of water absorbed per gram of dry flour. OAC was determined using the method of Lin et al. [26]. A 0.5 g of flour was thoroughly mixed with 6 mL of corn oil (Fula, Algés, Portugal) in a 15 mL centrifuge tube. After allowing the mixture to rest for 30 min, it was centrifuged at 3000× *g* for 25 min at 4 °C. The supernatant was discarded, and OAC was calculated as the grams of oil absorbed per gram of dry matter. The WAI, WSI, and SP were determined using a modified method of Abebe et al. [27]. A 2.5 g flour sample was placed in a 50 mL centrifuge tube, mixed with 30 mL of distilled water, and heated in a water bath at 90 °C for 15 min. After cooling to room temperature, the tubes were centrifuged at 3000× *g* for 10 min at 4 °C. The supernatant was then transferred to a pre-weighed evaporating dish to determine the solid content while the sediment was weighed. The WAI was calculated as the grams of sediment per gram of flour. To determine the WSI, the solids were recovered by evaporating the supernatant for 16 h at 105 °C, and the WSI was expressed as the weight of the recovered solids divided by the weight of the flour, multiplied by 100 to obtain the percentage of the soluble material present in the flour. SP was calculated by dividing the weight of the sediment obtained after centrifugation by the difference between the flour sample’s initial weight and the dry solids recovered from the supernatant after evaporation.

### 2.4. Amylose, Amylopectin and Damaged Starch Content

The amylose and amylopectin contents were determined using the Megazyme commercial kit (Wicklow, Ireland), (REF: K-AMYL) from Wicklow, Ireland, while the damaged starch content was analyzed using the Megazyme kit (Wicklow, Ireland), (REF: K-SDAM). All experiments were performed in triplicate.

### 2.5. Thermal Properties by Differential Scanning Calorimetry (DSC)

The thermal properties of the flour samples were determined using a TA-60WS DSC (Shimadzu Corporation, Kyoto, Japan) differential scanning calorimeter calibrated by Indium. Approximately 3 mg of each flour sample was placed in a 40 µL high pressure aluminum hermetic cells (6 diameter × 5 mm—5 MPa) along with 9 mg of deionized water. The pans were sealed and kept at room temperature for 1 h to allow moisture equilibration. Scanning was performed from 30 °C to 180 °C at a heating rate of 10 °C/min, under a nitrogen flow rate of 40 mL/min. An empty sealed pan served as the reference. Gelatinization properties, including enthalpy of gelatinization (ΔH_gel_, J/g of flour), peak temperature (T_p-gel_), onset temperature (T_o-gel_), and conclusion temperature (T_c-gel_), were determined from the thermal scans. Each sample was analyzed in triplicate.

### 2.6. Morphological Evaluation by Scanning Electron Microscopy (SEM)

A scanning electron microscope (Phenom XL G2, Thermo Fisher Scientific, Waltham, MA, USA), equipped with an X-ray detector, was used to observe the morphology of starch granules and capture detailed images of their microstructure. The samples were placed on observation pins with double-sided adhesive carbon tape (NEM tape, Nisshin, Japan) and then coated with a thin layer of gold/palladium using a sputter coater (Polaron, Bad Schwalbach, Germany) for enhanced imaging. Imaging was performed in high vacuum/conductivity mode without pre-metallization, using an accelerating voltage of 5 kV. The micrographs were captured using a secondary electron detector at magnifications of 500×, 1500×, 2500×, and 3500×.

### 2.7. X-Ray Diffraction (XRD)

The X-ray diffraction patterns of the samples were obtained using a MiniFlex 600 diffractometer (Rigaku, Tokyo, Japan) with Cu-Kα radiation (λ = 0.154 nm) at 40 kV and 15 mA. The measurements were conducted at room temperature, covering a 2θ range of 3° to 70° with a step size of 0.01° and a scan rate of 3° per minute. All tests were performed in triplicate, and the relative crystallinity percentage (RC) was determined by calculating the ratio of the crystalline area ($C_{a}$) to the amorphous area ($A_{a}$) using Equation (1), with calculations performed using OriginPro software (Version 2022. OriginLab Corporation, Northampton, MA, USA).(1)$$RC\;\left(\%\right)=\left(\frac{C_{a}}{C_{a}+A_{a}}\right)\times 100$$

### 2.8. Fourier Transform Infrared (FTIR) Spectroscopy

Fourier Transform Infrared (FTIR) spectra were collected using a PerkinElmer Spectrum BX FTIR System equipped with a DTGS detector (Waltham, MA, USA) in triplicate. Spectra were recorded in diffuse reflectance mode with a PIKE Technologies Gladi attenuated total reflectance (ATR) accessory. Measurements were conducted over a wavenumber range of 4000–400 cm−^1^ with a resolution of 4 cm−^1^. The ATR crystal was thoroughly cleaned between each sample, and a new background spectrum was acquired using the empty crystal prior to each measurement.

### 2.9. Rheological Measurement

The rheological properties of the samples were evaluated using dynamic oscillatory tests on a controlled-stress rheometer (model CS-50, Bohlin Instruments, Cranbury, NJ, USA) equipped with a cone-plate geometry (40 mm diameter, 0.15 mm gap). To determine the appropriate sample preparation, the least gelatinization concentration (LGC) was first identified, following the method of Kaushal et al. [28]. In this procedure, flour concentrations of 2%, 4%, 6%, 8%, 10%, 12%, 14%, 16%, 18%, and 20% were mixed with distilled water for total volume of 5 mL in glass tubes. The mixtures were vortexed for 20 s to ensure proper dispersion of the solids, then cooked in a water bath at 100 °C for one hour. After cooking, the samples were immediately cooled in an ice bath and refrigerated for 2 h. The LGC was determined as the lowest concentration at which the gel remained intact without slipping when the test tube was inverted. This concentration was subsequently used for the rheological analysis.

Rheological properties were assessed using both amplitude sweep and frequency sweep tests. The amplitude sweep was conducted with a strain range of 0.01 to 10 at a fixed frequency of 1 Hz, with the aim of determining the linear viscoelastic region (LVR). The frequency sweep test was then performed within the LVR, using a frequency range of 1 to 10 Hz, based on the strain values identified in the amplitude sweep test (0.1 for all samples). Each sample was measured in triplicate, and each replicate was analyzed three times, resulting in a total of 9 measurements per sample variation.

The power law model fitted the obtained data (Equations (2) and (3)) as described by [29].(2)$$G^{\prime }=G_{0}^{\prime }.\omega^{n^{\prime }}$$(3)$$G^{\prime \prime }=G_{0}^{\prime \prime }.\omega^{n^{\prime \prime }}$$

$G^{\prime }$ and $G^{\prime \prime }$ represent the elastic and viscous moduli, respectively, while the coefficients $G_{0}^{\prime }$ and $G_{0}^{\prime \prime }$ correspond to the elastic and viscous moduli at a frequency of 1 Hz. The exponents $n^{\prime }$ and $n^{\prime \prime }$ describe the dependence of these moduli on the oscillation frequency (ω).

### 2.10. Statistical Analysis

All measurements were performed at least in triplicate. Statistical analysis was performed using SPSS software (version SPSS 28.0, IBM Corp., Armonk, NY, USA). A one-way analysis of variance (ANOVA) was conducted to determine the statistical significance of the differences between sample means. The Shapiro–Wilk test was used to evaluate data normality, while Levene’s test assessed the homogeneity of variances. If the assumption of homogeneity of variances was violated, Welch’s ANOVA was employed as a robust alternative to standard ANOVA. Post hoc comparisons were performed using Tukey’s HSD for standard ANOVA or Games–Howell for Welch’s ANOVA to identify significant differences at a significance level of *p* < 0.05. Nonlinear least-squares regression modeling in rheology analysis was performed using SPSS. Results are reported as mean ± standard deviation.

## 3. Results and Discussion

### 3.1. Techno-Functional Properties

Table 1 presents the flour samples’ techno-functional properties, including WAC, OAC, WAI, WSI, and SP. Water absorption capacity (WAC) is primarily determined by the hydrophilic groups in proteins and carbohydrates within the flour matrix, which influence water retention. Gluten-free flours generally absorb more water in formulations than wheat-based doughs [30]. In this study, ultrasound treatment significantly decreased WAC. It has been observed that cavitation effects from ultrasound reduced the hydrophilicity of these compounds [31]. A similar reduction in WAC was also noted in sonicated rice flour treated at room temperature [31]. The decrease in water absorption capacity was more pronounced in the 15% sonicated flour than the 25% sonicated sample, likely due to the more substantial effect of sonication at lower flour concentrations in water. This reduction may result in lower dough viscosity, enhancing dough handling and machinability but potentially reducing water retention during baking, which could affect bread volume and texture [32].

Oil absorption capacity (OAC) is defined as the grams of lipid absorbed per gram of flour. This value is crucial in the development of new products, as lipids act as flavor retainers within the matrix and can enhance the mouthfeel of the final product [28]. In this study, sonication significantly increased the OAC, with the highest values observed in the 15% sonicated flour. A similar increase has been reported in sonicated oat starch [33] and quinoa protein isolate [34]. This enhancement in OAC may result from the exposure of hydrophobic protein regions to the external environment and the entrapment of oil molecules within pores and fissures created by ultrasound treatment [33]. The increased oil absorption capacity indicates structural modifications that could enhance interactions with lipids and emulsifiers, potentially improving dough stability and crumb softness [35].

The water absorption index (WAI) indicates the capacity of flour or starch to retain water under extreme conditions, such as mixing and baking. This value is expressed in grams of water per gram of dry flour and is crucial for achieving a stable texture in the final product, as a higher WAI helps improve dough cohesion and texture, which is particularly important for gluten-free bread. In this study, sonicated flours exhibited lower WAI values than native flour, likely due to the physical and chemical disruption of compounds by cavitation, resulting in decreased water uptake.

The water solubility index (WSI) indicates the content of soluble compounds within the flour matrix, which can differ across different flours depending on both the treatment conditions and the intrinsic properties of their components. In this study, native flour obtained 0.8 g/100 g solubility, while sonicated flours showed a significant increase in WSI (6.5 and 5.6 g/100 g, respectively, for 15% and 25%). Previous studies [36,37] suggest that sonicated starch samples often show more significant increases in water solubility than flour samples. This difference is likely due to the presence of other components in flour, particularly proteins, which may restrict this effect. Since corn starch constitutes a significant portion of this composite gluten-free flour, a notable increase in solubility was observed after sonication. Furthermore, ultrasonication (US) can enhance starch solubility by releasing amylose chains, which have lower structural integrity than amylopectin and are more susceptible to ultrasonic disruption [24]. Similar effects have been noted in studies on sonicated quinoa and corn flours [38]; however, in another study on brown and white teff flours [39], solubility was observed to decrease under higher treatment temperatures. The increase in solubility is likely due to cavitation effects disrupting the amorphous and crystalline regions of starch granules, increasing porosity without fundamentally breaking down amylose and amylopectin content (results observed in amylose, amylopectin, and damaged starch content—Table 1), which makes granules more accessible to water without altering their internal amylose content.

The swelling power (SP) represents the extent to which starch granules in the flour absorb water and swell. In this study, the swelling power of sonicated flours decreased significantly compared to the non-sonicated sample. During sonication, cavitation disrupts the crystalline structure of starch granules, particularly affecting the hydrogen bonding within amylopectin. This disruption leads to a breakdown of the amylopectin’s organized, water-retaining network, reducing its capacity to swell. Additionally, since sonication can promote the exposure of amylose and amylose-lipid complexes, these components can further inhibit swelling in the treated samples [40].

### 3.2. Amylose, Amylopectin, and Damaged Starch Content

Table 1 presents the results of this study for amylose, amylopectin, and damaged starch content. It was observed that sonication, independently of the flour concentration used, did not cause any significant changes in the damaged starch content compared to the control sample. This suggests that applied sonication conditions may not have been sufficient to induce noticeable mechanical or thermal damage to the starch granules, as typically observed in other studies employing more intense treatment parameters.

Significant changes were observed in the amylose and amylopectin content of the sonicated samples. Specifically, the amylose content significantly increased compared to the control sample. This increase can be attributed to the cavitation effect produced by ultrasound. Cavitation involves microbubbles’ rapid formation, growth, and collapse in the liquid medium, generating localized high-energy zones. These zones can disrupt the granular structure of starch and facilitate the rupture of α-1,6-glucosidic bonds within the side chains of amylopectin. The fracture of these bonds results in the release of amylose molecules, thereby increasing the amylose content in the sonicated samples and correspondingly decreasing amylopectin content [39]. A higher amylose content can contribute to firmer crumb structure and improved retrogradation behavior, benefiting shelf-life stability [41].

### 3.3. Thermal Properties by DSC

Table 2 presents the thermal properties of the analyzed flours. The gelatinization thermograms refer to the order–disorder transition phase of granules when starch/flour is heated in excess of water [42]. The thermograms from both sonicated and non-sonicated composite gluten-free flours show a single endothermic peak associated with starch granule gelatinization. Gelatinization temperatures and enthalpy of gelatinization (ΔH_gel_), reflecting the energy needed to disrupt hydrogen bonds or double helices in crystalline and amorphous starch regions [43], were recorded. Three key temperatures characterize the stages of gelatinization: the onset temperature (T_o-gel_), where the weakest crystallites begin to melt; the peak temperature (T_p-gel_), representing the maximum energy absorption and disruption of double helices; and the conclusion temperature (T_c-gel_), indicating the final melting of the strongest crystalline bonds, marking gelatinization completion. These properties are important in indicating starch quality and can provide basic information regarding the different processing applications of starches or flours [44]. 

Results in Tabe 2 showed that the enthalpy of gelatinization decreased in the sonicated samples compared to non-sonicated flour. This reduction can be explained by the effects of cavitation, which refers to the rapid formation and collapse of bubbles in the liquid, facilitating starch gelatinization during baking and potentially improving loaf volume. This process releases energy that breaks down the double helices in starch’s crystalline and non-crystalline regions, weakening the structural order [45]. This disruption of the amorphous regions allows water molecules to penetrate the starch granules more easily and reach the crystalline zones [46]. This results in less energy needed for gelatinization because water can interact more readily with the starch structures, especially where cavitation-induced flaws and channels form in the granules (as shown in SEM images—Figure 1) [47]. The same result was observed in sonicated white and brown teff flour [48], quinoa flour [49], rice flour [31], corn and pea starch [50], and oat starch [33]. The temperature of conclusion (Tc-gel) increased significantly in sonicated flours in comparison with the non-sonicated sample. This increase in the conclusion temperature could be due to sonication enhancing the interactions between starch and protein components in the composite flour, such as potato protein and chickpea flour, which may contribute to a more stable gel network in the baked product. These strengthened interactions contribute to a more stable network structure, requiring a higher temperature to complete gelatinization [51]. On the other hand, no significant difference was observed in the onset and peak temperatures of gelatinization between sonicated and non-sonicated gluten-free composite flours. Since these temperatures reflect the initial melting of crystalline regions, the unchanged To-gel and Tp-gel suggest that sonication did not significantly disrupt these structures, leaving the energy needed to start gelatinization stable.

### 3.4. Scanning Electron Microscopy (SEM)

Figure 1 presents a comparison of the morphology between the non-sonicated and sonicated (15% and 25% flour concentration) gluten-free composite flour as observed through a Scanning Electron Microscopy (SEM). The energetic vibrations from ultrasound waves cause cavitation bubbles to burst, creating tiny regions of extremely high pressure and fast-moving streams. These combined effects can degrade starch granules. The non-sonicated sample displays particles with a noticeably larger and more uniform size distribution. In contrast, both sonicated samples exhibit a wider range of particle sizes, with evident fracturing in some particles, resulting in smaller fragments. The non-sonicated sample reveals a predominance of rounded or elliptical particles, indicative of their natural state. However, the sonicated samples show a greater prevalence of irregular or angular shapes. This observation aligns with the theory that fracturing from the sonication process disrupts the original particle shape, leading to sharper edges and uneven surfaces [52].

Surface texture also presents a clear distinction between the samples. The non-sonicated sample exhibits a smooth surface texture associated with minimal disruption of the flour particle surface. In contrast, the sonicated samples reveal a rougher and more irregular surface texture.

SEM micrographs strongly suggest that the cavitation effects generated by ultrasound impact the surface characteristics and morphology of flour particles. This translates to a decrease in the overall particle size due to fracturing, an increase in the irregularity of particle shape, and a roughening of the surface texture compared to the non-sonicated control.

### 3.5. X-Ray Diffraction (XRD) Analysis

Figure 2 shows the studied gluten-free composite samples’ X-ray diffraction pattern and relative crystallinity (RC). In this formulation, rice flour and corn starch are the primary components, resulting in a diffractogram where A-type starch is the dominant form across all samples. The characteristic diffraction peaks at 15.2°, 17.21°, 18.47°, 23.14°, and 26.63° indicate the presence of A-type starch contributed by rice flour, maize starch, chickpea flour, and pregelatinized rice flour. Additionally, B-type starch, primarily from potato starch, is indicated by peaks at 5.5°, 14.5°, 17.5°, 22.1°, and 23.9°. Weak peaks at higher scattering angles correspond to salt content in the composite flour.

In the sonicated samples (15% and 25% flour concentration), certain weak peaks decreased in intensity compared to the non-sonicated control, also suggesting that ultrasound waves penetrated the starch structure, causing partial degradation. The primary mechanism by which ultrasonication affects starch crystallinity is the selective degradation of amorphous regions, which are less densely packed and, therefore, more susceptible to ultrasonic energy than the highly ordered crystalline regions [31,37,53]. The calculated RC values show a significant decrease in crystallinity in the sonicated samples, indicating that ultrasound cavitation was able to penetrate the compact crystalline structure of the flour, causing some disruption in both the crystalline and amorphous regions. This reduction in crystallinity aligns with the observed decrease in the intensity of specific peaks in the X-ray diffractogram.

### 3.6. FTIR Analysis

The FTIR analysis was used to evaluate the structural modifications in short-range/double helical order of starch induced by sonication in the flours. In this analysis, the absorption bands at 1047, 1022, and 995 cm^−1^ were studied to identify the changes in the starch structure caused by sonication (Table 3). These bands are associated with the absorbance at crystalline structure, amorphous structure, and C-OH bending vibrations, respectively [54]. The absorption ratios 1047/1022 and 1022/995 were used to evaluate changes in the crystalline and amorphous regions as well as the organization of double helices within the crystallites [55]. The 1047/1022 ratio, which quantifies the degree of short-range order crystallinity, shows a decrease in sonicated samples compared to the non-sonicated sample, but this decrease was not statistically significant. This suggests that while sonication may cause minor disruptions in the localized crystalline arrangement of starch granules, these changes were not substantial enough to be detected by FTIR in terms of short-range molecular order [56]. In contrast, an XRD analysis (Section 3.5) revealed a significant decrease in overall crystallinity in the sonicated samples, indicating that sonication had a more pronounced effect on the long-range crystalline structure of the starch. These subtle changes indicate that the ultrasound treatment primarily acts on the physical structure of the starch without inducing significant structural disorganization. The lack of new or lost absorption peaks further supports this, showing that no chemical bond formation or breakage occurred, confirming that sonication is a purely physical modification process [57,58,59,60].

The 1022/995 ratio, associated with the organization of double helices within the crystalline regions, showed a more distinct trend. While the non-sonicated and 15% sonicated samples were not significantly different, and the 15% and 25% sonicated samples were also similar, the 25% sonicated sample exhibited a significant decrease in this ratio compared to the non-sonicated sample. This finding indicates that higher sonication intensities may induce more significant disruption to the helical organization within the crystallites, potentially reflecting partial disorganization of the double helices. The mechanical effects of sonication, such as cavitation-induced shear forces and turbulence, likely play a role in these structural changes, particularly at higher flour concentrations [61].

### 3.7. Rheological Measurements

The rheological properties of the gels derived from the studied flours were analyzed using oscillatory tests, as shown in Figure 3, with the power-law model parameters detailed in Table 4. Strain sweep tests identified the linear viscoelastic region (LVR) and the cross-over point ($G^{\prime }$ = $G^{\prime \prime }$), where the gel transitions from elastic to viscous-dominant behavior. Frequency sweep tests performed at a constant strain of 0.1% across a frequency range of 1–10 Hz showed that the elastic modulus ($G^{\prime }$) was consistently higher than the viscous modulus ($G^{\prime \prime }$) for all samples, indicating solid-like behavior and classifying the systems as true gels [62]. This effect can be explained by the cavitation-induced breakdown of starch granules, which form hydrogen bonds with water molecules, leading to a more elastic network [24]. The same results were observed in the studies of sonicated corn starch [37], corn and pea starch [50], white and brown teff flour [39], and rice flour [63] where $G^{\prime }$ is higher than $G^{\prime \prime }$. The elastic modulus ($G^{\prime }$) of sonicated samples was higher than that of the non-sonicated control, with the 15% hydrated sample exhibiting the most pronounced increase. This suggests that sonication enhances the structural rigidity and elasticity of the gel network, possibly due to the rearrangements made to the enhanced interactions of amylose and amylopectin or compaction of molecular components under moderate concentration conditions [64]. In contrast, the viscous modulus ($G^{\prime \prime }$) was higher in the non-sonicated sample compared to the sonicated ones, indicating that sonication reduces the dissipative (viscous) behavior of the gel, likely by disrupting weaker interactions and enhancing elastic contributions.

The complex viscosity (η*) results, shown in Figure 3, illustrate the shear-thinning behavior of all samples, as η* decreased with the increasing frequency. This behavior is typical of gel systems, where structural rearrangements occur under applied stress, reducing resistance to flow [65]. The 15% sonicated sample exhibited slightly higher η* values than the non-sonicated and 25% sonicated samples at lower frequencies, suggesting that sonication at this concentration level enhances the gel’s structural integrity and energy dissipation capabilities. This indicates a stronger network capable of resisting deformation under low-frequency oscillatory stress. Conversely, the 25% sonicated sample consistently showed the lowest η* across the frequency range, pointing to a significant weakening of the gel network under excessive flour concentration conditions. The reduction in η* in the 25% concentrated sample suggests disrupted molecular interactions and reduced network cohesiveness caused by overconcentration during sonication.

The power-law model parameters further complement these observations by providing quantitative insights into the rheological parameters. The fitted parameters, $G_{0}^{\prime }$ and $G_{0}^{\prime \prime },\;$ shown in Table 4, are used to reflect the gel’s viscous and elastic behavior, respectively; the $n^{\prime }$ and $n^{\prime \prime }$ value parameters are used to describe the gel’s frequency dependence. For $G^{\prime \prime }$, the 15% sonicated sample exhibited the highest $G_{0}^{\prime \prime }$ value, which, although not significantly different from the non-sonicated control, indicates enhanced viscous contributions under these conditions. On the other hand, the frequency exponent ($n^{\prime \prime }$) for $G^{\prime \prime }$ in the 15% sonicated sample was significantly lower than in the other samples, reflecting a reduced sensitivity to frequency changes and a more stable gel structure. Regarding $G^{\prime }$, the 25% sonicated sample showed a lower frequency exponent ($n^{\prime }$), implying a diminished dependency on the frequency and reduced elastic stability compared to the other samples. These results collectively indicate that while moderate sonication (15% concentration) improves the gel’s viscous response and structural integrity, excessive flour concentration (25%) weakens both elastic and viscous contributions, diminishing the overall gel stability and functionality.

## 4. Conclusions

This study demonstrates that an ultrasound treatment significantly modifies a gluten-free composite flour’s functional, structural, thermal, and rheological properties, with potential benefits for breadmaking. Sonication, particularly at the 15% flour concentration, reduced the water absorption capacity and swelling power, likely due to a decreased starch concentration, which could enhance dough handling but affect water retention during baking. Increased oil absorption and solubility suggest improved interactions with lipids, potentially enhancing dough stability and crumb softness. Structural changes, including an increased amylose content and reduced crystallinity, indicate improved gel formation and retrogradation behavior, which could contribute to better texture and shelf-life. Thermal analysis revealed a decrease in gelatinization enthalpy, suggesting weakened starch crystallinity and faster gelatinization, which may improve loaf volume. Rheological results showed a stronger, more elastic dough network, which is crucial for gluten-free systems where structure formation is challenging. These findings suggest that the ultrasound can be a promising tool for enhancing gluten-free flour functionality and improving bread quality. However, further baking trials are necessary to confirm its impact on final product attributes such as texture, sensory properties, and staling behavior. Additionally, optimizing sonication conditions could allow for a better control over gluten-free dough properties, leading to higher quality baked goods.

## Figures and Tables

**Figure 1 foods-14-01094-f001:**
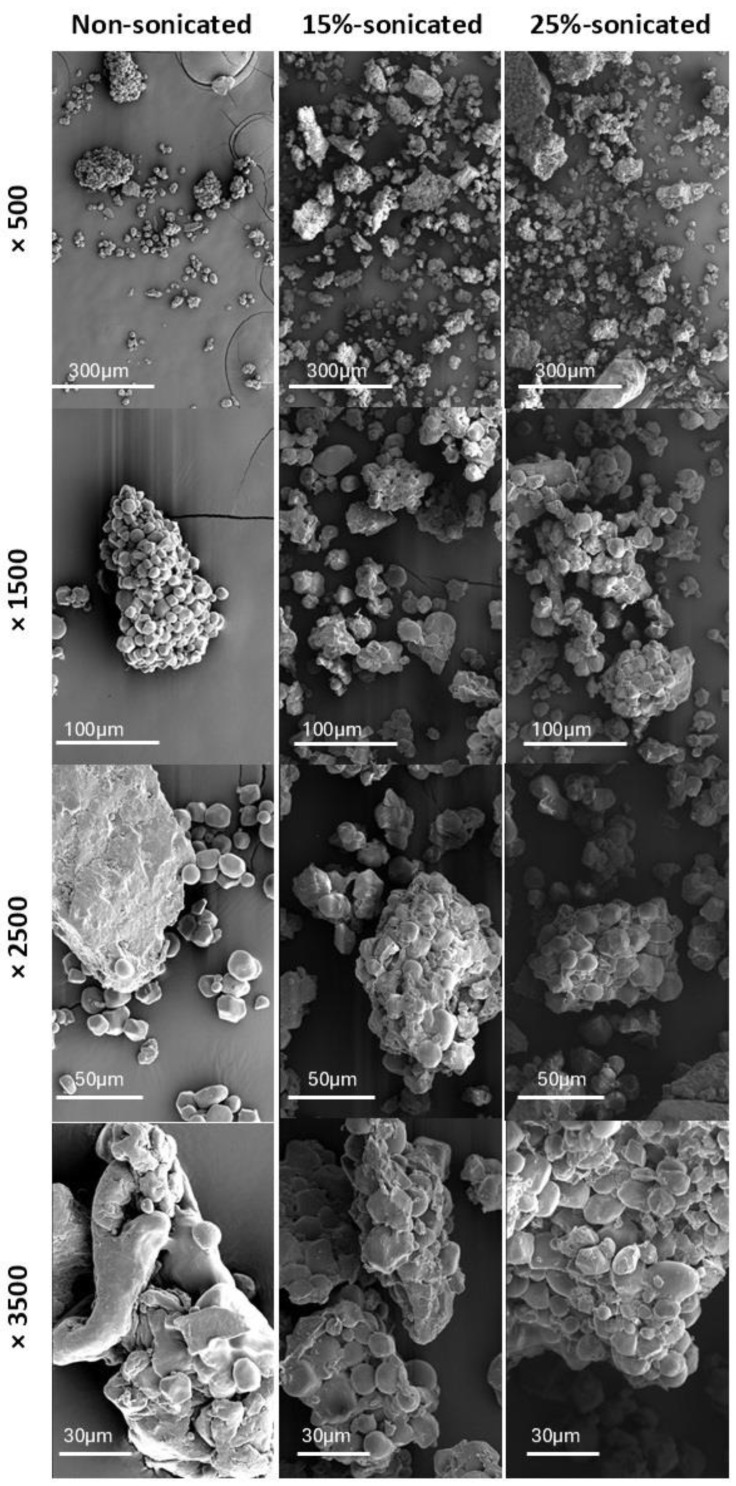
SEM micrographs of non-sonicated and sonicated (15% and 25% flour concentration) gluten-free composite flour (500×, 1500×, 2500×, and 3500×).

**Figure 2 foods-14-01094-f002:**
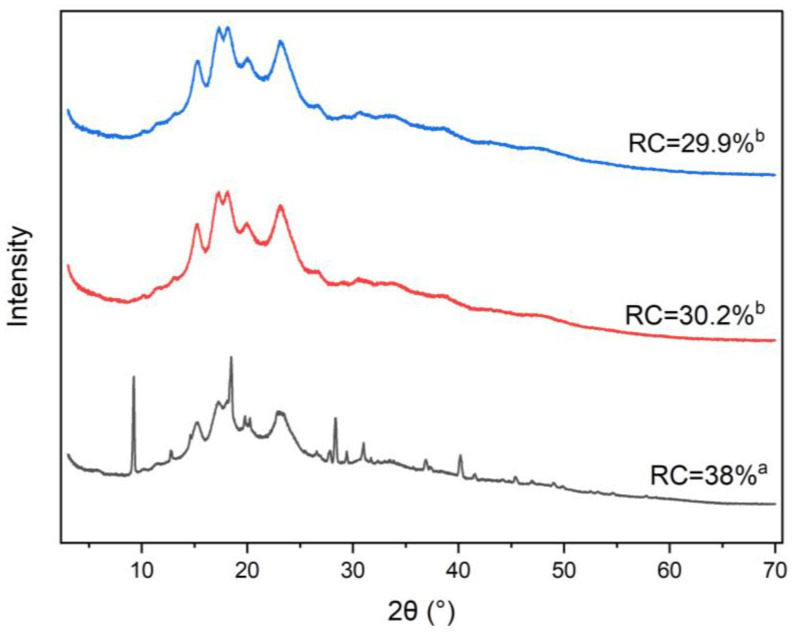
X-ray diffraction pattern and relative crystallinity (RC) of the studied gluten-free composite flours. Blue pattern: 25%-sonicated, red pattern: 15%-sonicated, black pattern: Non-sonicated. Different letters indicate a significant difference between the sample’s mean values (*p* < 0.05).

**Figure 3 foods-14-01094-f003:**
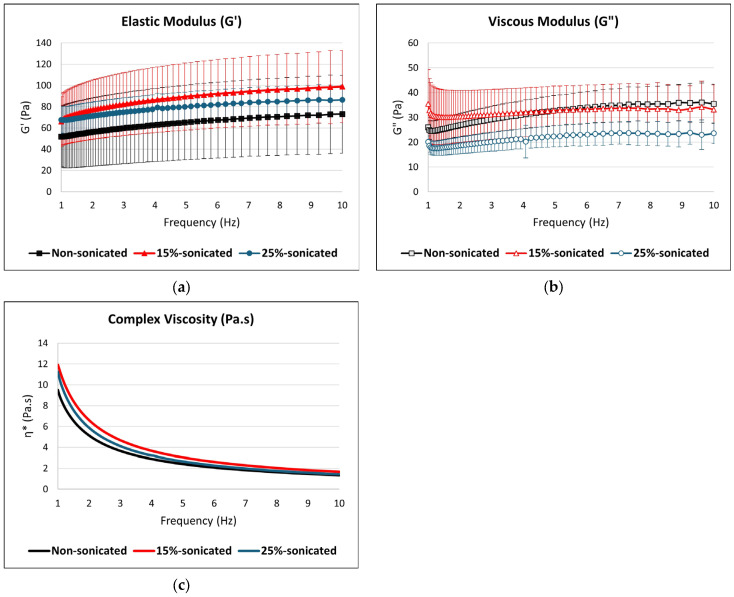
Rheological parameters: (**a**) elastic modulus ($G^{\prime }$), (**b**) viscous modulus ($G^{\prime \prime }$), and (**c**) complex viscosity (η*) of the sonicated and non-sonicated gluten-free composite flour.

**Table 1 foods-14-01094-t001:** Functional properties, amylose, amylopectin, and damaged starch content of studied flours.

Sample	WAC (g/g)	OAC (g/g)	WAI (g/g)	WSI (g/100 g)	SP (g/g)	Damaged Starch Content (%)	Amylose Content (%)	Amylopectin Content (%)
**Non-sonicated**	1.81 ± 0.05 ^a^	0.94 ± 0.02 ^a^	12.54 ± 0.16 ^a^	0.8 ± 0.0 ^a^	12.6 ± 0.2 ^a^	12.0 ± 0.6 ^a^	23.5 ± 0.5 ^a^	76.5 ± 0.5 ^a^
**15%-sonicated**	0.87 ± 0.02 ^b^	1.38 ± 0.06 ^b^	7.38 ± 0.26 ^b^	6.5 ± 0.2 ^b^	7.9 ± 0.3 ^b^	12.2 ± 0.4 ^a^	38.5 ± 0.6 ^b^	61.5 ± 0.7 ^b^
**25%-sonicated**	1.23 ± 0.06 ^c^	1.24 ± 0.06 ^c^	8.29 ± 0.23 ^c^	5.6 ± 0.5 ^c^	8.8 ± 0.2 ^c^	11.9 ± 0.1 ^a^	47 ± 8 ^b^	52 ± 8 ^b^

Statistically significant differences between the sample’s mean property at *p* < 0.05 are indicated by different letters within the same column.

**Table 2 foods-14-01094-t002:** Thermal properties related to the studied flours.

Sample	T_o-gel_ (°C)	T_p-gel_ (°C)	T_c-gel_ (°C)	ΔH_gel_ (J/g)
**Non-sonicated**	68.5 ± 1.5 ^a^	71.5 ± 1 ^a^	78 ± 2 ^a^	−1.3 ± 0.1 ^a^
**15%-sonicated**	68.9 ± 1.1 ^a^	72.3 ± 0.4 ^a^	82.3 ± 1.1 ^b^	−1.0 ± 0.0 ^b^
**25%-sonicated**	68.5 ± 0.4 ^a^	74.6 ± 2 ^a^	81.7 ± 0.7 ^b^	−1.1 ± 0.1 ^b^

Statistically significant differences between the sample’s mean property at *p* < 0.05 are indicated by different letters within the same column.

**Table 3 foods-14-01094-t003:** Starch structure FTIR analysis on control and treated flours.

	Non-Sonicated	15%-Sonicated	25%-Sonicated
**1047/1022**	1.29 ± 0.03 ^a^	1.27 ± 0.03 ^a^	1.25 ± 0.02 ^a^
**1022/995**	1.15 ± 0.02 ^a^	1.11 ± 0.01 ^ab^	1.11 ± 0.01 ^b^

Different letters in the row indicate statistically significant differences between each sample mean at *p* < 0.05.

**Table 4 foods-14-01094-t004:** Power law model parameters of gels rheological properties.

Sample	$G_{0}^{\prime }$(Pa)	$n^{\prime }$	$G_{0}^{\prime \prime }$(Pa)	$n^{\prime \prime }$
**Non-sonicated**	51 ± 30 ^a^	0.18 ± 0.05 ^a^	24 ± 4 ^a^	0.19 ± 0.03 ^a^
**15%-sonicated**	69 ± 25 ^a^	0.16 ± 0.00 ^a^	30 ± 11 ^ab^	0.06 ± 0.07 ^b^
**25%-sonicated**	66 ± 13 ^a^	0.11 ± 0.00 ^b^	17 ± 3 ^ac^	0.15 ± 0.02 ^a^

Different letters within the same column indicate statistically significant differences between the sample’s mean rheological parameter at *p* < 0.05.

## Data Availability

The original contributions presented in this study are included in the article/Appendix A. Further inquiries can be directed to the corresponding author.

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
