# Peer review of "Impact of Ultrasound on a Gluten-Free Composite Flour Based on Rice Flour and Corn Starch for Breadmaking Applications"

_foods, 2025, doi:10.3390/foods14071094_

Round 1

Reviewer 1 Report

Comments and Suggestions for Authors

The development and research of gluten-free products is a hot topic in the food industry. This manuscript systematically studied the effects of ultrasound treatment on the technological, functional, thermal, structural, and rheological properties of composite gluten-free protein flour at two different hydration concentrations. The research results demonstrate the potential of ultrasound to improve gluten-free protein flour, aiding in the development of gluten-free products. The manuscript is well-written, with comprehensive research content, standardized figures and tables, correct results analysis, and thorough discussion. Here are some comments:

  1. The relevant research on the effects of ultrasound and other treatment methods on flour characteristics in the introduction needs to be strengthened, and the statements regarding gluten-free flour can be simplified.
  2. The conclusion needs to be simplified, and many details should be included in the discussion section within the results analysis.
  3. How was the process condition of ultrasound treatment determined? Were preliminary experiments conducted or were there references?

4. Lines 92-95 state that the total composition of the mixed flour is 100.05%, please verify.

Author Response

We sincerely appreciate the reviewers’ valuable comments, suggestions, and recommendations. We have provided a detailed point-by-point reply below, where our response and revised part follow each comment (italicized).

The development and research of gluten-free products is a hot topic in the food industry. This manuscript systematically studied the effects of ultrasound treatment on the technological, functional, thermal, structural, and rheological properties of composite gluten-free protein flour at two different hydration concentrations. The research results demonstrate the potential of ultrasound to improve gluten-free protein flour, aiding in the development of gluten-free products. The manuscript is well-written, with comprehensive research content, standardized figures and tables, correct results analysis, and thorough discussion. Here are some comments:

Comment1: The relevant research on the effects of ultrasound and other treatment methods on flour characteristics in the introduction needs to be strengthened, and the statements regarding gluten-free flour can be simplified.

Response: We sincerely appreciate the reviewer’s insightful comments and suggestions; we have revised the introduction to incorporate more relevant literature while simplifying certain sections as suggested.

Revised Part: “Ultrasound is a promising physical modification technique that alters the functional and structural properties of flour. Previous studies have demonstrated its impact on water absorption, solubility, swelling behavior, and rheological characteristics, making it a useful tool for improving gluten-free formulations [17]. Compared to conventional methods such as enzymatic or hydrothermal treatments, ultrasound offers a sustainable, non-thermal alternative with minimal chemical alterations, making it a green, safe, and easily applicable technology that has gained significant attention. [18,19]

AND

Formulating and optimizing gluten-free systems requires analysing factors like texture, nutritional content, and sensory qualities to ensure high product quality and functionality.

Comment2: The conclusion needs to be simplified, and many details should be included in the discussion section within the results analysis.

Response: Thank you for your feedback. We have simplified the conclusion by removing detailed findings and relocating them to the discussion section within the result analysis. The revised conclusion now provides a concise summary of the key outcomes and their implications.

Revised Part: This study demonstrates that ultrasound treatment significantly modifies a gluten-free composite flour's functional, structural, thermal, and rheological properties, with potential benefits for breadmaking. Sonication, particularly at 15% hydration, reduced water absorption capacity and swelling power, likely due to decreased starch hydration, which may enhance dough handling but affect water retention during baking. Increased oil absorption and solubility suggest improved interactions with lipids, potentially enhancing dough stability and crumb softness. Structural changes, including increased amylose content and reduced crystallinity, indicate improved gel formation and retrogradation behavior, which could contribute to better texture and shelf-life. Thermal analysis revealed a decrease in gelatinization enthalpy, suggesting weakened starch crystallinity and faster gelatinization, which may improve loaf volume. Rheological results showed a stronger, more elastic dough network, which is crucial for gluten-free systems where structure formation is challenging. These findings suggest ultrasound can be a promising tool for enhancing gluten-free flour functionality and improving bread quality. However, further baking trials are necessary to confirm its impact on final product attributes such as texture, sensory properties, and staling behavior. Additionally, optimizing sonication conditions could allow for better control over gluten-free dough properties, leading to higher-quality baked goods.

Comment 3. How was the process condition of ultrasound treatment determined? Were preliminary experiments conducted or were there references?

Response: The ultrasound treatment conditions were selected based on literature references, as there are various conditions used for studying the effects of ultrasound on flours and starches. In the first step of our study, we aimed to investigate the impact of different hydration concentrations of the sample under sonication. Therefore, we followed conditions commonly used in similar studies to ensure comparability while focusing on hydration effects. Preliminary experiments were not conducted, as our objective was to first observe the structural and functional modifications under these conditions before optimizing parameters in future studies. A short explanation was added to clarify why these conditions were chosen for the study.

Revised: These sonication conditions were selected based on the study by Farrokhi et al.,[24] which reviewed and compared the effects of various sonication conditions on flours and starches. The most commonly used and effective conditions reported in the literature were chosen for this research.

Comment 4: Lines 92-95 state that the total composition of the mixed flour is 100.05%, please verify.

Response: Thank you for bringing this to our attention. The total composition of the mixed flour was previously reported as 100.05% due to rounding adjustments in the amounts of the corn starch, rice flour and pregelatinized rice flour. We have now revised the composition to reflect the precise amounts, ensuring a total of exactly100%.

Revised: This blended flour consists primarily of corn starch and rice flour (49.11% and 23.86%, respectively), along with other ingredients like pregelatinized rice flour (8.58%), potato starch (5.6%), dextrose (4.9%), chickpea flour (2.2%), potato protein (2%), salt (1.5%), xanthan gum (1%), HPMC (Hydroxypropyl Methyl Cellulose) (1%), transglutaminase (0.2%), and carob flour (0.05%).

Reviewer 2 Report

Comments and Suggestions for Authors

This well-written manuscript holds significant implications for the properties of gluten-free flour modified using ultrasound. However, there are still the following issues:

  1. The hydration levels of 15% and 25% are somewhat confusing; based on the experimental method, they refer to the concentration of the samples in the reaction system, so it is recommended to use simpler and more understandable terminology.
  2. Line 103 should clarify whether 15% and 25% refer to a mass/volume ratio or a mass/mass ratio.
  3. In Table 1, for Germen non-sonicated, Germen 15%-sonicated, and Germen 25%-sonicated, the authors should explain them in the main text and the figure caption to avoid confusion.
  4. Similarly, in Figure 1, different samples are named non-sonicated, 15%-sonicated, and 25%-sonicated. Is this nomenclature different from that in Table 1?
  5. Please explain why there was no control group of natural flour + ultrasound treatment.
  6. In lines 108-109, why were the samples not directly dried to the target moisture content after ultrasound treatment?
  7. It is recommended to change the title “Rice Flour and Corn Starch” to “composite gluten-free flour” because, according to the content, these two ingredients have not been studied separately but have consistently existed in a mixed form. The authors only need to clarify their composition in the Materials section.

Author Response

We sincerely appreciate the reviewers’ valuable comments, suggestions, and recommendations. We have provided a detailed point-by-point reply below, where our response and revised part follow each comment (italicized).

This well-written manuscript holds significant implications for the properties of gluten-free flour modified using ultrasound. However, there are still the following issues:

Comment 1: The hydration levels of 15% and 25% are somewhat confusing; based on the experimental method, they refer to the concentration of the samples in the reaction system, so it is recommended to use simpler and more understandable terminology.

Response: Thank you for the reviewer's valuable feedback. The reviewer is correct that the terms "15%" and "25%" refer to the concentration of the flour in the reaction system, specifically the amount of flour relative to the total amount of water and flour used during sonication. To clarify, we revised the terminology in the manuscript to reflect the concentration more clearly.

Revised: Instead of "hydration levels/concentrations," we refer to these as "flour concentrations (15% and 25%)" in all paper to make it more straightforward and easily understandable.

Comment 2: Line 103 should clarify whether 15% and 25% refer to a mass/volume ratio or a mass/mass ratio.

Response: Thank you for pointing this out. To clarify, the 15% and 25% refer to a mass/mass (w/w) ratio of flour to total amount (flour and water) used in the sonication process.

Revised: We updated in the revised manuscript for greater clarity.

Comment 3: In Table 1, for Germen non-sonicated, Germen 15%-sonicated, and Germen 25%-sonicated, the authors should explain them in the main text and the figure caption to avoid confusion.

Response: Thank you for your comment. To avoid confusion, we have ensured consistency by referring to all samples (Non-sonicated, 15%-sonicated, and 25%-sonicated) in the same manner throughout the manuscript, both in the main text and the figure captions.

Revised: Samples were identified as Non-sonicated (control, without sonication), 15%-sonication (sonicated with 15% flour concentration), and 25%-sonication (sonicated with 25% flour concentration).

Comment 4: Similarly, in Figure 1, different samples are named non-sonicated, 15%-sonicated, and 25%-sonicated. Is this nomenclature different from that in Table 1?

Response and Revised: Thank you for your comment. We have revised the text and ensured that the nomenclature is uniform across Table 1 and Figure 1 to maintain consistency.

Comment 5: Please explain why there was no control group of natural flour + ultrasound treatment.

Response: Thank you for your comment. The study does include a control sample, which is the natural (non-sonicated) composite flour. The properties of the sonicated samples were compared against this non-sonicated control to evaluate the effects of ultrasound treatment. To avoid confusion, we have added the sample identification in the Materials and Methods section.

Comment 6: In lines 108-109, why were the samples not directly dried to the target moisture content after ultrasound treatment?

Response: Thank you for your question. The samples were not directly dried to the target moisture content after ultrasound treatment to ensure uniform moisture distribution and prevent rapid drying, which could affect the structural and functional properties of the flour. This approach helps maintain consistency in the analysis.

Comment 7: It is recommended to change the title “Rice Flour and Corn Starch” to “composite gluten-free flour” because, according to the content, these two ingredients have not been studied separately but have consistently existed in a mixed form. The authors only need to clarify their composition in the Materials section.

Response: Thank you for your suggestion. The title explicitly states, "based on Rice Flour and Corn Starch", which indicates that the study focuses on a composite gluten-free flour rather than analysing these ingredients separately. Since "composite gluten-free flour" is a broad term, specifying its key components in the title provides more clarity for readers. However, to further ensure there is no confusion, we have added a more detailed explanation of the flour composition in the Materials and Methods section. This should make it clear that the study investigates the effects of ultrasound on a pre-formulated gluten-free flour blend rather than individual ingredients.

Reviewer 3 Report

Comments and Suggestions for Authors

The study could benefit from a broader range of ultrasound parameters (e.g., different power intensities, frequencies, or treatment durations) to fully map the modification effects.
Increasing the number of replicates or providing more detailed statistical analyses (e.g., confidence intervals or effect sizes) would enhance the robustness of the findings.

The conclusions are well aligned with the evidence presented. The observed decrease in water absorption capacity and swelling power, coupled with increased oil absorption and water solubility, supports the claim that ultrasound treatment modifies the functional behavior of the flour. Thermal analysis showing a lower gelatinization enthalpy—along with minimal changes in crystallinity—suggests that the structural modifications are localized (likely in the amorphous regions), which is consistent with the SEM and FTIR findings.
The tables are well organized and clearly present quantitative data on particle size, hydration properties, thermal properties, and rheological behavior.
Figures (such as SEM images, FTIR spectra, and rheological curves) effectively illustrate the microstructural changes and functional modifications.
One suggestion would be to ensure that error bars and statistical markers (like significance letters) are clearly legible in all figures, so readers can easily appreciate the variability and significance of the changes observed.

Overall, this study is a valuable contribution to the field of gluten‐free food technology, providing both fundamental insights and practical guidance for developing improved ingredients.

Comments on the Quality of English Language

The English could be improved to more clearly express the research in the Introduction.

Author Response

We sincerely appreciate the reviewers’ valuable comments, suggestions, and recommendations. We have provided a detailed point-by-point reply below, where our response and revised part follow each comment (italicized).

The study could benefit from a broader range of ultrasound parameters (e.g., different power intensities, frequencies, or treatment durations) to fully map the modification effects.
Increasing the number of replicates or providing more detailed statistical analyses (e.g., confidence intervals or effect sizes) would enhance the robustness of the findings.

The conclusions are well aligned with the evidence presented. The observed decrease in water absorption capacity and swelling power, coupled with increased oil absorption and water solubility, supports the claim that ultrasound treatment modifies the functional behavior of the flour. Thermal analysis showing a lower gelatinization enthalpy—along with minimal changes in crystallinity—suggests that the structural modifications are localized (likely in the amorphous regions), which is consistent with the SEM and FTIR findings.

The tables are well organized and clearly present quantitative data on particle size, hydration properties, thermal properties, and rheological behavior.

Figures (such as SEM images, FTIR spectra, and rheological curves) effectively illustrate the microstructural changes and functional modifications.

One suggestion would be to ensure that error bars and statistical markers (like significance letters) are clearly legible in all figures, so readers can easily appreciate the variability and significance of the changes observed.

Overall, this study is a valuable contribution to the field of gluten‐free food technology, providing both fundamental insights and practical guidance for developing improved ingredients.

Response: Thank you for your valuable feedback and thoughtful suggestions. We appreciate your recommendation to explore a broader range of ultrasound parameters. While our study focused on specific conditions to evaluate their impact, we acknowledge that a more comprehensive investigation with varying power intensities, frequencies, or treatment durations could further map the modification effects. We will consider this in future research.

Regarding statistical analysis, we ensured rigorous experimental design with appropriate replicates and statistical tests to validate our findings.

We are pleased that our conclusions align with the presented evidence and that the tables and figures effectively communicate the key findings. We appreciate your suggestion regarding the clarity of error bars and significance markers in the figures. We have reviewed them and made necessary adjustments to ensure they are clearly visible and easily interpretable.

Thank you again for your insightful review and for recognizing the contribution of our study to gluten-free food technology. We believe these revisions will further enhance the clarity and impact of our work.

Reviewer 4 Report

Comments and Suggestions for Authors

As a reviewer, I have gone through the manuscript entitled “Impact of Ultrasound on a Gluten-free Composite Flour based on Rice Flour and Corn Starch for Breadmaking Applications”. The aim of this paper is to investigate the impact of sonication on the techno-functional, thermal, structural, and rheological properties of a composite gluten-free flour. The work is very well structured, presents investigations regarding amylose, amylopectin and damaged starch content, functional properties (water absorption capacity, oil absorption capacity, water absorption index, water solubility index, swelling power), thermal properties (differential scanning calorimetry), morphological evaluation (scanning electron microscopy), X-ray diffraction analysis, infrared spectroscopy (FTIR), rheological measurement and statistical analysis. The concepts are clearly presented and can be easily understood by interested readers. However, it exists some points (minor) that need to be addressed (please, see them below detailed point-by-point) to improve the scientific quality of the submitted manuscript paper:

  • To improve the quality of the work and increase the interest of the readers, I recommend the short description of other treatment methods. In the introduction part of the manuscript it was said that it only exists (line 68-70);
  • Ftir analysis toghther with X-ray diffraction is a good method for structural analysis. I recommend putting the Ftir spectra in text or at least in a separate file (supplementary information)

Author Response

We sincerely appreciate the reviewers’ valuable comments, suggestions, and recommendations. We have provided a detailed point-by-point reply below, where our response and revised part follow each comment (italicized).

As a reviewer, I have gone through the manuscript entitled “Impact of Ultrasound on a Gluten-free Composite Flour based on Rice Flour and Corn Starch for Breadmaking Applications”. The aim of this paper is to investigate the impact of sonication on the techno-functional, thermal, structural, and rheological properties of a composite gluten-free flour. The work is very well structured, presents investigations regarding amylose, amylopectin and damaged starch content, functional properties (water absorption capacity, oil absorption capacity, water absorption index, water solubility index, swelling power), thermal properties (differential scanning calorimetry), morphological evaluation (scanning electron microscopy), X-ray diffraction analysis, infrared spectroscopy (FTIR), rheological measurement and statistical analysis. The concepts are clearly presented and can be easily understood by interested readers. However, it exists some points (minor) that need to be addressed (please, see them below detailed point-by-point) to improve the scientific quality of the submitted manuscript paper:

  • To improve the quality of the work and increase the interest of the readers, I recommend the short description of other treatment methods. In the introduction part of the manuscript, it was said that it only exists (line 68-70);
  • FTIR analysis together with X-ray diffraction is a good method for structural analysis. I recommend putting the FTIR spectra in text or at least in a separate file (supplementary information)

Response: Thank you very much for your positive feedback and constructive comments. We appreciate your suggestions to improve the manuscript.

1. Regarding the description of other treatment methods:

We agree that providing a brief overview of other treatment methods in the introduction could enhance the reader's understanding and broaden the context of the study. We added a short description of other relevant treatment methods in the introduction to highlight their significance. This will give readers a clearer background on the topic.

Revised: The food industry employs various modification techniques, such as genetic, mechanical, physical, chemical, or enzymatic methods, to alter the natural physicochemical properties of gluten-free ingredients, enabling them to better meet specific processing needs [16,17]. Modifying raw GF materials before dough preparation is crucial for improving bread quality, as starch granule characteristics significantly impact the final product, due to their interactions with other components in the bread formulation [18]. Among the different modification techniques, physical processes like ultrasonication (US), high hydrostatic pressure (HHP), microwave (MW), extrusion, ozone treatment, drum drying (DD), heat-moisture treatment (HMT), and annealing (ANN) are favored in the food industry for their environmental benefits and ability to produce 'clean label' products.

2. Regarding the FTIR spectra and X-ray diffraction analysis:

Thank you for your suggestion to include the FTIR spectra. We agree that including them would strengthen the manuscript’s clarity. To maintain the flow of the main text, we will move the FTIR spectra to the supplementary information section and mention this in the main text. The X-ray diffraction results will remain as presented in the manuscript, as we believe they are critical for understanding the structural modifications.

Once again, thank you for your thoughtful comments. These revisions will certainly improve the quality and clarity of the manuscript.